

# Violence, runaway, and suicide attempts among people living with schizophrenia in China: Prevalence and correlates

Yixiang Long[1], Xiaoliang Tong[1], Michael Awad[2], Shijun Xi[3] and Yu Yu[2,3]

[1] Department of Nursing, Third Xiangya Hospital, Central South University, Changsha, Hunan, China
[2] Division of Prevention and Community Research, Department of Psychiatry, Yale University School of Medicine, New Haven, CT, USA
[3] Department of Social Medicine and Health Management, Xiangya School of Public Health, Central South University, Changsha, Hunan, China

Corresponding author
Xiaoliang Tong, 601630@csu.edu.cn

## ABSTRACT

**Background:** People living with schizophrenia are at higher risk of disruptive behaviors, including violence, running away from home, and suicide attempts, which often co-occur and are highly correlated, yet seldom studied together. The current study investigated the frequency and correlates of disruptive behaviors among a Chinese community sample of individuals living with schizophrenia.

**Methods:** A cross-sectional study was conducted among 400 individuals living with schizophrenia from 12 communities. Data about disruptive behaviors in the past 2 months was collected using self-designed questionnaires. Clinical characteristics including psychiatric symptoms, depression, anxiety, disability, and functioning were collected by internationally standardized assessment instruments.

**Results:** About one-fifth (21%) of the subjects had experienced at least one form of disruptive behavior in the past 2 months. Violence was the most commonly reported (17.25%), which included damaging property (15%) and physical violence toward others (7.5%); followed by running away (6.5%), and suicide attempts (4%). Logistic regression analysis suggested that medication non-adherence (OR = 4.96, 95% CI [1.79–13.72]), involuntary hospital admission (OR = 5.35, 95% CI [2.06–13.87]), depression (OR = 2.34, 95% CI [1.07–5.10]), and lower social functioning (OR = 0.97, 95% CI [0.93–0.99]) were independently associated with a higher risk of disruptive behaviors.

**Conclusions:** The overlap among three forms of disruptive behaviors warrants them to be assessed and studied together in clinical, research, and policy fields. The significant association between disruptive behaviors with medication non-adherence, involuntary admission, depression, and lower social functioning indicates the need for integrated, targeted, and needs-based intervention programs to be developed for the prevention and treatment of these disruptive behaviors.

## INTRODUCTION

Schizophrenia is a debilitating, persistent psychiatric disorder characterized by symptoms such as delusions, hallucinations, and lack of emotional responsiveness and motivation, which significantly impair a person's cognition and social functioning (*APA, 1994*). Globally, people living with schizophrenia are at a higher risk of a range of disruptive behaviors, including physical violence (*Bottesi et al., 2021*; *Brown, O'Rourke & Schwannauer, 2019*; *Fazel et al., 2009*; *Slamanig et al., 2021*), running away from home (*Jaafari et al., 2019*), and suicide (*Ayalew, Defar & Reta, 2021*; *De Sousa, Shah & Shrivastava, 2020*; *Lyu, Zhang & Hennessy, 2021*; *Ran et al., 2020*; *Sher & Kahn, 2019*), than the general population. A pooled estimated prevalence of 18.5–35.4% for violence among schizophrenia individuals and other psychosis has been reported in various reviews (*Large & Nielssen, 2011*; *Li et al., 2020*; *Witt, van Dorn & Fazel, 2013*; *Zhou et al., 2016*), with risk estimated at seven times that of the general population in both retrospective and cohort studies (*Mullen et al., 2000*; *Tiihonen et al., 1997*). High rates of running away from home have also been reported among people with mental illness (*Jaafari et al., 2019*), although less studied among individuals living with schizophrenia. For suicide, a lifetime suicide rate of 5–10% has been reported among individuals living with schizophrenia by various reviews (*Hor & Taylor, 2010*; *Sher & Kahn, 2019*), with standardized mortality ratios between 10 and 20 (*Hor & Taylor, 2010*).

Although violence, running away, and suicide have been viewed as distinct disruptive behaviors, accumulating evidence has demonstrated that they often co-occur and are highly correlated with each other (*Meltzer et al., 2012*; *Witt, Hawton & Fazel, 2014*). However, research has focused on these behaviors separately, and rates and risk factors for any disruptive behaviors among individuals living with schizophrenia are rarely reported in the same study. Several reviews have summarized a range of risk factors for violence among individuals living with schizophrenia, including socio-demographics, past history, treatment-related factors, psychiatric symptoms, mood disorders, substance abuse, *etc.* (*Rund, 2018*; *Witt, van Dorn & Fazel, 2013*). Similar risk factors have also been reported for runaway and suicide attempts among individuals living with schizophrenia (*Jaafari et al., 2019*; *Sher & Kahn, 2019*; *Shlafer, Poehlmann & Donelan-McCall, 2012*). In light of the overlapping risk factors shared by violence, running away, and suicide, it is thus both interesting and important to take into account all three forms of disruptive behaviors simultaneously to gain a more integrated and inclusive picture of disruptive behaviors among individuals living with schizophrenia.

In China, increasing attention has been paid to disruptive behaviors of people with serious mental illness, with several national policies, regulations, and programs initiated successively by the Chinese government. In 2004, the Chinese government started the 686 Program with one important goal being identifying and treating people with serious mental illness who are at high risk of disruptive behaviors (*Ma, 2012*). In 2011, the Chinese government launched its first National System of Basic Information Collection and Analysis for Psychoses mainly to collect data on their violent or socially-disruptive behaviors (*Zhou & Xiao, 2015*). In 2012, China passed its first National Mental Health Law

allowing for involuntary admission of people with serious mental illness who are highly violent (*Shao & Xie, 2013*). In 2016, the Chinese government issued a Reward Policy to financially incentivize and reward family management of disruptive behaviors of persons with serious mental illness (*Yu, Zhou & Xiao, 2018*).

Although great political attention has been directed towards disruptive behaviors of serious mental illness in the community, little research has been done in this area. Most of the existing research on this topic has been hospital-based (*Zhou et al., 2016*). Since the majority of individuals with schizophrenia are living in the community, community-based research on disruptive behaviors is warranted. In addition, disruptive behaviors including violence, running away, and suicide have been mostly studied separately both in China and abroad, although increasing evidence has shown high overlaps among these behaviors and their risk factors (*Jaafari et al., 2019*; *Sher & Kahn, 2019*; *Shlafer, Poehlmann & Donelan-McCall, 2012*; *Witt, Hawton & Fazel, 2014*). There is a lack of research simultaneously studying the three forms of disruptive behaviors in the same sample. In response to these limitations, we conducted the current study to investigate the prevalence and risk factors of three common forms of disruptive behaviors: violence, running away, and suicide in a community sample of individuals living with schizophrenia in China.

# MATERIALS AND METHODS

## Study population

This cross-sectional study was conducted in Changsha Psychiatric Hospital from May 2019 to September 2019 as a baseline assessment of a WeChat-based family intervention project focused on improving the family burden of people living with schizophrenia in China (*Yu et al., 2020a*). The hospital provides community-based mental health care in Changsha City through the 686 Program, China's largest demonstration project in mental health to provide free mental health care for people with serious mental illness (*Ma, 2012*). A two-stage cluster-sampling method was adopted to identify participants. In the first stage, we randomly selected the following four districts out of the nine administrative districts of Changsha City: Gaoxin District, Furong District, Kaifu District, and Tianxin District. In the second stage, we randomly selected 2–4 communities from each district based on population size and the registered number of clients in the 686 program. Our final sampling frame is all 1,068 registered clients with serious mental illness from the 12 communities that were representative of Changsha city. The inclusion criteria included: (1) registered in the 686 Program; (2) diagnosed as schizophrenia by the Chinese Classification of Mental Disorders-3 (CCMD-3) or the International Classification of Diseases-10 (ICD-10); (3) aged 18 and older; (4) living with at least one family member; (5) able to read and verbally-communicate; (6) visited the community health center for each monthly free medicine refill during the study periods. The exclusion criteria included: (1) had a diagnosis other than schizophrenia, (2) absent for five monthly medicine refills at the community health center during the 5-month study period. Based on the exclusion criteria, 618 clients were excluded, leading to 450 eligible clients. Among the 450 clients approached for study participation, 50 didn't complete the study due to refusal

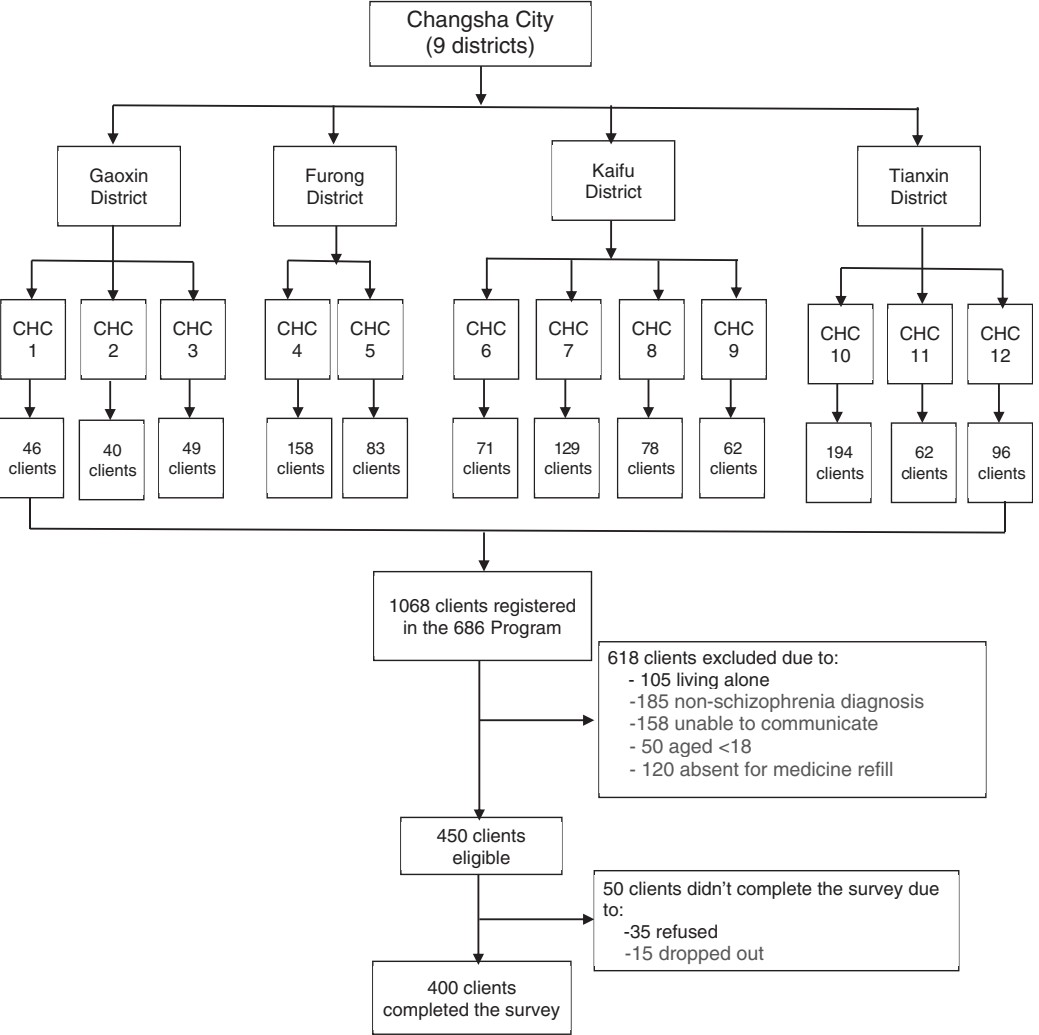

**Figure 1 Flowchart of subject enrollment.** CHC, Community health center, full name of the 12 CHCs are listed as below: CHC1, Gaoxin District Dongfanghong Town Health Center; CHC2, Gaoxin District Leifeng Street Community Health Service Center; CHC3, Gaoxin District Liao Jiaping Nursing Home; CHC4, Furong District Parity Hospital; CHC5, Furong District Red Cross Hospital; CHC6, Kaifu District Laodaohe Union Hospital; CHC7, Kaifu District Geriatric Hospital; CHC8, Kaifu District Hongshan Street Community Health Service Center; CHC9, Kaifu District Shaping Street Community Health Service Center; CHC10, Tianxin District Jinpenling Community Health Service Center; CHC11, Tianxin District Datuo Town Health Center; CHC12, Tianxin District Pozi Street Community Health Service Center.               

of participation or dropping out during the study, leading to a final sample of 400 clients with a response rate of 88.89%. Figure 1 shows the flow chart of participant enrollment.

Each month, a team of at least three psychiatrists from the Changsha Psychiatric Hospital circulated the 12 communities to distribute free medication and run routine check-ups. Our research team followed the psychiatry team during their monthly community visits and approached potential participants when they came for medication refills. The research team explained the details of the study purpose and procedures and conducted face-to-face interviews with the participants after obtaining written informed

consent from them. In addition, the psychiatrist team assessed the symptoms and functioning of each participant using standardized assessment instruments. Each interview took approximately 20–30 min, and each participant was reimbursed with 10 Renminbi ($1.40 in 2019 dollars) after the interview. The study was reviewed and approved by the Institutional Review Board of the Xiangya School of Public Health, Central South University (No.: XYGW-2019-029).

## Instruments

### Sample characteristics

*Socio*-demographic information including gender, age, marital status, education, and employment were collected by a self-designed personal information sheet. In addition, we collected clinical characteristics such as medication non-adherence and involuntary admission in the past 6 months. Since most of the registered clients in the 686 Program followed the doctor's advice and took medications on a daily basis (*Yu et al., 2017*), we defined medication non-adherence as "not taking medicine on a daily basis each month". Involuntary admission was defined as "being sent to the hospital by the family members due to concerns of potential violence risk the client may have when the client herself/himself doesn't agree to be hospitalized." Disruptive behaviors and other clinical characteristics were measured using scales which are described as below:

### Disruptive behaviors

Disruptive behaviors including violence, running away, and suicide attempts were assessed by asking whether the participants had the following four behaviors in the past 2 months: damaging properties (violence), physical violence (violence), suicide attempts, and running away from home. Damaging properties were defined as any forms of aggressive behaviors towards objects, such as breaking windows, smashing up furniture, and other forms of property destruction. Physical violence was defined as any form of aggressive behavior toward other people such as scratching, punching, and hitting people. Suicide attempts were defined as any form of non-fatal, self-directed, potentially injurious behavior with an intent to die. Running away from home was defined as escaping the family caregiver's custody and leaving home without advance notice and permission from the family members, thus leading to family to look for them everywhere.

### Psychiatric symptoms

Psychiatric symptoms were assessed by the 18-item Brief Psychiatric Rating Scale (BPRS-18) that includes five domains of clinical symptoms: affect, positive symptoms, negative symptoms, resistance, and activation (*Shafer, 2005*). The total score ranges from 0–126, with a higher score representing more severe psychiatric symptoms. The BPRS-18 showed good internal consistency with a Cronbach's alpha of 0.85 in the current study.

### Depression

Depression was assessed by the nine-item Patient Health Questionnaire-9 (PHQ-9) for screening of depressive symptoms in the past 2 weeks (*Spitzer, Kroenke & Williams, 1999*). The total score ranges from 0 to 27, with a higher score indicating more depressive

symptoms, and a cut-off point of 10 differentiating depression and non-depression (*Manea, Gilbody & McMillan, 2015*). The PHQ-9 showed good internal consistency with a Cronbach's alpha of 0.92 in the current study.

### Anxiety

Anxiety was assessed by the seven-item Generalized Anxiety Disorder Scale-7 (GAD-7) for screening of anxiety symptoms in the past 2 weeks (*Schalet et al., 2014*). The total score ranges from 0 to 21, with a higher score indicating more anxiety symptoms, and a cut-off point of 10 differentiating anxiety and non-anxiety (*Schalet et al., 2014*). The GAD-7 showed good internal consistency with a Cronbach's alpha of 0.96 in the current study.

### Disability

Disability was assessed by the 12-item World Health Organization Disability Assessment Schedule 2.0 (WHODAS 2.0) for disability and functional impairment (*WHO, 2017*). The total score ranges from 0–48, with a higher score representing a higher level of disability. The WHODAS 2.0 showed good internal consistency (Cronbach's alpha of 0.89) in the current study.

### Functioning

Functioning was assessed by the Global Assessment of Functioning (GAF) for psychological, social, and occupational functioning. It is a one-item scale measured on a hypothetical continuum of functionality ranging from 1 to 100, with a higher score indicating better patient functioning (*Goldman, Skodol & Lave, 1992*).

## Statistical analysis

Exploratory and summary statistics were obtained for all variables within the dataset. Socio-demographic and clinical characteristics were compared between the disruptive group and non-disruptive group by $\chi^2$ test for categorical variables or *t*-test for continuous variables. A multivariate logistic regression was further conducted to determine correlates of disruptive behaviors. All data were analyzed using STATA version 16.

## RESULTS

### Prevalence of disruptive behaviors

Table 1 shows the prevalence of each disruptive behavior committed by the participants. Approximately one fifth (21.0%) had engaged in at least one form of disruptive behavior in the past 2 months, with damaging properties being the most commonly reported (15.0%), followed by physical violence toward other people (7.5%), running away from home (6.5%), and suicide attempts (4.0%). The prevalence of violence was 17.25% when combining damaging properties and attacking people. Among the disruptive group, most had only engaged in one disruptive behavior (76.2%), while less than one-third (23.8%) had engaged in two concurrent disruptive behaviors. A further comparison of disruptive behaviors by gender showed non-significant differences.

**Table 1 Prevalence of disruptive behaviors among schizophrenia individuals ($n = 400$).**

| Variables | Total ($n = 400$) | | Male ($n = 200$) | | Female ($n = 200$) | | P |
|---|---|---|---|---|---|---|---|
| | N | % | N | % | N | % | |
| Violence | 69 | 17.25 | 35 | 17.5 | 34 | 17.0 | 0.895 |
| Damaging property | 60 | 15.0 | 28 | 14.0 | 32 | 16.0 | 0.575 |
| Attacking people | 30 | 7.5 | 17 | 8.5 | 13 | 6.5 | 0.448 |
| Run away | 26 | 6.5 | 14 | 7.0 | 12 | 6.0 | 0.685 |
| Suicide attempt | 16 | 4.0 | 9 | 4.5 | 7 | 3.5 | 0.610 |
| Any disruptive behavior | 84 | 21.0 | 42 | 21.0 | 42 | 21.0 | 0.644 |
| Only one form of behavior | 64 | 76.2 | 31 | 73.8 | 33 | 78.6 | 0.490 |
| Two forms of behavior | 13 | 15.5 | 6 | 14.3 | 7 | 16.7 | |
| All three forms of behavior | 7 | 8.3 | 5 | 11.9 | 2 | 4.7 | |

## Comparison between disruptive and non-disruptive group

Table 2 shows the socio-demographic and clinical characteristics of the participants and their comparisons between the disruptive and non-disruptive groups. In general, the sample corresponds to a profile of middle-aged (mean age: $46.87 \pm 10.99$), married (43.00%), and unemployed (89.50%) man/woman (50% each), with middle and high school education (67.75%). A further comparison of socio-demographic characteristics between the disruptive and non-disruptive groups demonstrated no significant differences except for employment, where the disruptive group had a higher unemployment rate than the non-disruptive group (96.43% *vs* 87.66%, $p = 0.020$).

With regards to clinical characteristics, only 7.20% had medication non-adherence, and 8.25% reported involuntary hospital admission in the past 2 months. Emotional distress was relatively common among the participants, with 40.51% and 29.74% screened positive for depression and anxiety, respectively. The total sample had a mean score of $32.90 \pm 11.43$ for psychiatric symptoms, $26.02 \pm 10.22$ for disability, and $61.83 \pm 13.58$ for functioning. A further comparison of clinical characteristics between disruptive and non-disruptive groups showed significant differences in each of the measured characteristics. Compared to the non-disruptive group, the disruptive group had higher rates of medication non-adherence (15.48% *vs* 4.92%, $p = 0.001$), involuntary admission (20.24% *vs* 5.06%, $p < 0.01$), depression (67.07% *vs* 33.44%, $p < 0.01$) and anxiety (53.01% *vs* 23.45%, $p < 0.01$). The disruptive group also showed more severe psychiatric symptoms ($40.12 \pm 13.58$ *vs* $30.99 \pm 9.97$, $p < 0.01$), higher rates of disability ($30.54 \pm 10.67$ *vs* $24.77 \pm 9.74$, $p < 0.01$), and lower levels of functioning ($53.03 \pm 14.05$ *vs* $64.21 \pm 12.45$, $p < 0.01$) than the non-disruptive group.

## Correlates of disruptive behaviors

Table 3 shows a multivariate logistic regression for correlates of disruptive behaviors, with disruptive behavior as the dependent variable and all socio-demographic and clinical characteristics as the independent variables. Among the 12 factors that were included in

**Table 2 Socio demographic and clinical characteristics by disruptive behaviors.**

| Socio-demographics | Total (*n* = 400) | | Non-disruptive group (*n* = 316) | | Disruptive group (*n* = 84) | | P |
|---|---|---|---|---|---|---|---|
| | Mean/*N* | SD/% | Mean/*N* | SD/% | Mean/*N* | SD/% | |
| Age | 46.87 | 10.99 | 46.81 | 10.64 | 47.07 | 12.27 | 0.847 |
| Gender | | | | | | | |
| Male | 200 | 50 | 158 | 50.00 | 42 | 50 | 0.644 |
| Female | 200 | 50 | 158 | 50.00 | 42 | 50 | |
| Marriage | | | | | | | |
| Single | 150 | 37.50 | 114 | 36.08 | 36 | 42.86 | 0.522 |
| Married/cohabited | 172 | 43.00 | 139 | 43.99 | 33 | 39.29 | |
| Else[a] | 78 | 19.50 | 63 | 19.94 | 15 | 17.86 | |
| Education | | | | | | | |
| Primary & below | 75 | 18.75 | 54 | 17.09 | 21 | 25 | 0.221 |
| Middle & high | 271 | 67.75 | 217 | 68.67 | 54 | 64.29 | |
| College & above | 54 | 13.50 | 45 | 14.24 | 9 | 10.71 | |
| Employment | | | | | | | |
| Unemployed | 358 | 89.50 | 277 | 87.66 | 81 | 96.43 | **0.020** |
| Employed | 42 | 10.50 | 39 | 12.34 | 3 | 3.57 | |
| **Clinical characteristics** | | | | | | | |
| Medication non-adherence | | | | | | | |
| No | 361 | 92.80 | 290 | 95.08 | 71 | 84.52 | **0.001** |
| Yes | 28 | 7.20 | 15 | 4.92 | 13 | 15.48 | |
| Involuntary admission | | | | | | | |
| No | 367 | 91.75 | 300 | 94.94 | 67 | 79.76 | **<0.001** |
| Yes | 33 | 8.25 | 16 | 5.06 | 17 | 20.24 | |
| Symptoms (BPRS) | 32.9 | 11.43 | 30.99 | 9.97 | 40.12 | 13.58 | **<0.001** |
| Depression (PHQ-9) | | | | | | | |
| No | 232 | 59.49 | 205 | 66.56 | 27 | 32.93 | **<0.001** |
| Yes | 158 | 40.51 | 103 | 33.44 | 55 | 67.07 | |
| Anxiety (GAD-7) | | | | | | | |
| No | 274 | 70.26 | 235 | 76.55 | 39 | 46.99 | **<0.001** |
| Yes | 116 | 29.74 | 72 | 23.45 | 44 | 53.01 | |
| Disability (WHODAS) | 26.02 | 10.22 | 24.77 | 9.74 | 30.54 | 10.67 | **<0.001** |
| Functioning (GAF) | 61.83 | 13.58 | 64.21 | 12.45 | 53.03 | 14.05 | **<0.001** |

Notes:
BPRS, Brief Psychiatric Rating Scale; PHQ-9, Patient Health Questionnaire-9; GAD-7, Generalized Anxiety Disorder Scale-7, WHODAS, World Health Organization Disability Assessment Schedule; GAF, Global Assessment of Functioning.
[a] Else includes separated, divorced, and widowed.
*P* values in bold represents significant at 0.05 level.

the model, four factors remained significant after controlling for all the other factors: medication non-adherence, involuntary admissions, depression, and functioning. The risk of disruptive behavior increased by four times among those with medication non-adherence (*OR*: 4.96, 95% CI [1.79–13.72]), and involuntary admission (*OR*: 5.35,

**Table 3 Multivariate logistic regression for correlates of disruptive behaviors.**

| Variables | | OR | 95% CI | P |
|---|---|---|---|---|
| **Socio-demographic characteristics** | | | | |
| Age | | 1.01 | 0.98–1.04 | 0.554 |
| Gender | Male | ref | | |
| | Female | 1.02 | 0.53–1.94 | 0.958 |
| Marriage | Single | ref | | |
| | Married/cohabited | 0.91 | 0.42–1.96 | 0.811 |
| | Else[a] | 0.68 | 0.26–1.76 | 0.423 |
| Education | Primary & below | ref | | |
| | Middle & high | 1.06 | 0.48–2.34 | 0.877 |
| | College & above | 1.41 | 0.44–4.58 | 0.565 |
| Employment | Unemployed | ref | | |
| | Employed | 0.74 | 0.20–2.77 | 0.651 |
| **Clinical characteristics** | | | | |
| Medication non-adherence | No | ref | | |
| | Yes | 4.96 | 1.79–13.72 | **0.002** |
| Involuntary admission | No | ref | | |
| | Yes | 5.35 | 2.06–13.87 | **0.001** |
| Symptoms (BPRS) | | 1.02 | 0.99–1.05 | 0.252 |
| Depression (PHQ-9) | No | ref | | |
| | Yes | 2.34 | 1.07–5.10 | **0.032** |
| Anxiety (GAD-7) | No | ref | | |
| | Yes | 1.34 | 0.61–2.95 | 0.466 |
| Disability (WHODAS) | | | | |
| Functioning (GAF) | | 0.97 | 0.93–0.99 | **0.006** |

**Notes:**

BPRS, Brief Psychiatric Rating Scale; PHQ-9, Patient Health Questionnaire-9; GAD-7, Generalized Anxiety Disorder Scale-7; WHODAS, World Health Organization Disability Assessment Schedule; GAF, Global Assessment of Functioning.

[a] Else includes separated, divorced, and widowed.

*P* values in bold represents significant at 0.05 level.

95% CI [2.06–13.87]) than their counterparts. Compared to those without depression, those with depression were over twice as likely to engage in disruptive behaviors (*OR*: 2.34, 95% CI [1.07–5.10]). On the other hand, every one-point increase in functioning decreased the risk of disruptive behavior by 3% (*OR*: 0.97, 95% CI [0.93–0.99]). No significant association was found between disruptive behavior with age, gender, marriage, education, employment, psychiatric symptoms, anxiety, or disability.

## DISCUSSION

To our knowledge, this is the first study to simultaneously examine three forms of disruptive behavior—violence, running away from home, and suicide attempts together using a representative community sample of individuals living with schizophrenia. The study showed 21% of individuals living with schizophrenia engaged in disruptive behaviors in the past 2 months, including 17.25% for violence (either damaging property

or physical violence toward others), 6.5% for running away, and 4.0% for suicide attempts. The prevalence of violence in the current study was comparable to the pooled prevalence of 18.5% reported by a review that included 45,533 individuals from 110 studies (*Witt, van Dorn & Fazel, 2013*). However, this violence rate was much lower than another review reporting a 34.5% rate of violence among first-episode psychosis. This discrepancy may be explained by the relatively long duration of illness (mean duration of 20 years) in the current sample as compared to the review samples who were experiencing their first episodes. This finding lends further support to the literature showing a greater risk of future psychosis among individuals experiencing first-episode psychosis than those in subsequent episodes (*Large & Nielssen, 2011*), indicating the need for early diagnosis and timely treatment for schizophrenia. The 17.25% prevalence rate for violence in the current study was also much lower than another meta-analysis focused on aggressive behaviors among individuals living with schizophrenia in China, which reported a pooled prevalence of 35.4% (95% CI [29.7–41.4]) (*Zhou et al., 2016*). This disparity may be best explained by the sample differences (community samples *vs* hospital samples), implying an increased risk of violence among individuals living with schizophrenia in a highly restrictive and controlled environment (*Zhou et al., 2016*).

Running away from home has been seldom studied among individuals living with schizophrenia in Western countries. This may be because many of them are adults who live alone instead of with their families. However, in Asian countries such as China, most individuals with schizophrenia live with their families, which is reflected in Confucian cultural values that the family should take the responsibility to live with and care for any sick member until they recover (*Yu et al., 2020b*). Having a family member with schizophrenia who runs away from home not only puts that individual at great risk of getting injured, engaging in unhealthy activities, becoming homeless, and even premature death (*Meltzer et al., 2012*), but also brings shame on their family members due to their lack of supervision and inadequate care of their loved one (*Yu et al., 2020b*). Our study, for the first time, found a 6.5% rate of running away from home among individuals living with schizophrenia, which warrants more scholarly attention and intervention in this area.

Regarding suicide attempts, inconsistent evidence exists on suicide attempts among people with schizophrenia. An earlier review by *Miles (1977)* showed a lifetime attempt rate of 10% for suicide associated with schizophrenia, which used to be the most cited figure by past studies. However, recent studies suggest 5% as a more representative suicide attempt rate associated with schizophrenia (*Hor & Taylor, 2010*). The 4.0% prevalence rate of suicide attempts in the current study also seems to support a lower suicide attempt, though this is still significantly higher than the reported 0.9% of the general population (*Hor & Taylor, 2010*). One implication of the current study is that risk for suicide should be included in the routine assessment of individuals living with schizophrenia, with more support and resources provided to those at higher risk.

The finding that medication non-adherence was associated with a four-fold increased risk of disruptive behavior was consistent with the bulk of studies. Non-adherence to medication among individuals living with schizophrenia has been well-established to be associated with a range of adverse outcomes, including violence, victimization, running

away, and suicide (*Appelbaum, 2019*; *Witt, van Dorn & Fazel, 2013*). It also greatly increased the severity of these disruptive behaviors and is associated with increased premature death (*Appelbaum, 2019*; *Witt, van Dorn & Fazel, 2013*). The mechanism underlying the positive association between medication non-adherence and disruptive behaviors may be explained by the exacerbation of psychiatric symptoms (especially positive symptoms) after discontinuation of medication, which, in turn, leads to a series of more disruptive behaviors such as violence and suicide (*Appelbaum, 2019*; *Witt, van Dorn & Fazel, 2013*). This finding underscores the need for early intervention and sustained engagement in treatment (*Buckley, 2012*).

Consistent with two reviews and meta-analyses showing a four-fold increase in violence risk associated with involuntary treatment (*Large & Nielssen, 2011*; *Witt, van Dorn & Fazel, 2013*), the current study found an even higher risk ratio of 5.35. Considering the highly-restrictive and controlled environment and usually long length of stay associated with hospitals, it is not surprising that individuals with histories of involuntary admission may have had more negative interpersonal experiences prior to or during these hospitalizations that may make them more likely to engage in disruptive behavior as a way of coping (*Zhou et al., 2016*). On the other hand, engaging in disruptive behaviors may have led to involuntary admission in the first place (*Large & Nielssen, 2011*). For instance, the first Chinese Mental Health Law stipulated that people with serious mental illness could be involuntarily hospitalized if they posed a risk to themselves or others (*Shao & Xie, 2013*). This finding represents an important implication for future studies to examine the dynamics between involuntary admission and disruptive behaviors. It also highlights the need to develop more community-based rehabilitation programs, complementary to involuntary hospitalization, in order to reduce future risk of disruptive behaviors and promote recovery among individuals living with schizophrenia in the community.

While robust evidence has consistently shown depression to be a strong predictor for both running away from home and attempting/completing suicide (*Tucker et al., 2011*), research on depression and violence have yielded divergent and conflicting conclusions, with some research showing negative associations (*Ekinci & Ekinci, 2013*) and some showing positive and even non-significant association (*Witt, van Dorn & Fazel, 2013*). Our findings support a positive association between depression and disruptive behaviors and indicate the need to treat depression in individuals living with schizophrenia. It is important to note that depressive symptoms have often been misunderstood as negative symptoms of schizophrenia and thus left undiagnosed and untreated among individuals with schizophrenia (*Xi et al., 2021*). Identifying depression and distinguishing depression from other schizophrenia symptoms should be the first key step towards decreasing depression and its related disruptive behaviors in individuals with schizophrenia.

Despite a large body of research reporting psychiatric symptoms as a major risk factor for disruptive behaviors in schizophrenia (*Rund, 2018*; *Witt, van Dorn & Fazel, 2013*), our current study failed to show such an association in multivariate regression. Instead, we found better functioning was significantly associated with decreased risk of disruptive behaviors, which has rarely been reported in previous literature. One explanation may be that psychiatric

symptoms and functioning are two overlapping constructs that are closely related to each other, the association between psychiatric symptoms and disruptive behaviors may be mediated by functioning. Further research is needed to explore such association and its underlying mechanism. This finding suggests improvement of functioning in schizophrenia may potentially serve an important purpose in the prevention of disruptive behaviors.

Several limitations should be acknowledged in the interpretation of the study findings. First, the sample was recruited from 12 communities in Changsha city of Hunan Province and may not represent people living with schizophrenia in other parts of China. Future national level and multi-center studies may be needed to provide a more comprehensive picture of disruptive behaviors among people living with schizophrenia in China. Second, the cross-sectional design of the study precludes any causal relationships between disruptive behaviors and risk factors, future longitudinal study designs are needed to establish causal relationships. Third, the prevalence of disruptive behaviors was only measured for a short time—the past 2 months—due to the concern of recall bias, which may be lower than the most commonly used time frame of 6 months or a lifetime. Future studies may consider using a longer time for disruptive behaviors assessment. Fourth, confirmation of disruptive behaviors relies solely on an individual's self-report, which may have been underreported by participants due to social desirability bias. Future studies may benefit from adding more objective assessment indicators such as an informant's description, medical and judicial records. Fifth, we didn't distinguish various types of schizophrenia diagnosis and thus were unable to compare disruptive behaviors by specific schizophrenia subtypes. Future studies may consider studying each specific type of schizophrenia and comparing the prevalence and risk factors of disruptive behaviors by schizophrenia diagnosis subtypes.

In spite of the above-mentioned limitations, our study still has some important innovations that may contribute to the literature. First, we studied three common forms of disruptive behaviors simultaneously in the same study, filling in the research gap of past studies that focused on only one form of disruptive behavior. Second, our sample was recruited from 12 communities in Changsha City and was representative of all community residents living with schizophrenia in Changsha City, which was also an understudied population compared to the most studied hospitalized population. Third, our results have identified four important and also modifiable risk factors for disruptive behaviors and provide useful information and guidance to inform future intervention and policies.

## CONCLUSION

Disruptive behaviors including violence, running away from home, and suicide are common among people living with schizophrenia in China, which calls for early recognition and intervention of these behaviors. The risk of disruptive behaviors was higher among those with medication non-adherence, involuntary hospital admission, depression, and lower social functioning, which provides implications for future targeted intervention programs to decrease the risk factors so as to mitigate disruptive behaviors.

## ACKNOWLEDGEMENTS

The authors would like to thank individuals for their participation in this study, family members who supported their participation, and staff from the Changsha Psychiatric Hospital and the 12 community health centers for their collaboration and support.

### Funding

This work was funded by a grant from the National Natural Science Foundation of China (Grant Number 71804197) in support of the senior author, Yu Yu. The funders had no role in study design, data collection and analysis, decision to publish, or preparation of the manuscript.

### Grant Disclosures

The following grant information was disclosed by the authors:
National Natural Science Foundation of China: 71804197.

### Competing Interests

The authors declare that they have no competing interests.

### Author Contributions

- Yixiang Long conceived and designed the experiments, performed the experiments, analyzed the data, prepared figures and/or tables, authored or reviewed drafts of the paper, and approved the final draft.
- Xiaoliang Tong conceived and designed the experiments, performed the experiments, analyzed the data, prepared figures and/or tables, authored or reviewed drafts of the paper, and approved the final draft.
- Michael Awad conceived and designed the experiments, prepared figures and/or tables, authored or reviewed drafts of the paper, and approved the final draft.
- Shijun Xi performed the experiments, authored or reviewed drafts of the paper, and approved the final draft.
- Yu Yu conceived and designed the experiments, performed the experiments, authored or reviewed drafts of the paper, and approved the final draft.

### Human Ethics

The following information was supplied relating to ethical approvals (*i.e.*, approving body and any reference numbers):

The study was reviewed and approved by the Institutional Review Board of the Xiangya School of Public Health, Central South University (No.: XYGW-2019-029).

### Data Availability

The raw data and codes are available in the Supplemental Files.

## Supplemental Information

Supplemental information for this article can be found online at http://dx.doi.org/10.7717/peerj.13033#supplemental-information.

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
