# Peer review of "Violence, runaway, and suicide attempts among people living with schizophrenia in China: Prevalence and correlates"

_PeerJ, doi:10.7717/peerj.13033_

## Round 0.1 · original submission · Major Revisions

Thanks for submitting your manuscript. This is a well-written work, however, it still needs some improvement before you want to publish it. The reasons for selecting China as the study location are not clearly justified. You mentioned a research gap; what exactly is the research gap in terms of broad literature review.?

In the methods section, kindly add a diagrammatic flow chart of the sample selection process.

·

Basic reporting

Grammar editing may be done. The title can be ' ,,,people living with schizophrenia' , instead of '...people living schizophrenia' . 'With' might have been left out from the title inadvertently. Grammar edit will improve the clarity of the observations presented in the paper. .

Experimental design

' no comment'

Validity of the findings

no comment'

Additional comments

This is a relevant area and the methodology is good. Editing for the English language and grammar will improve the quality of the paper.

·

Basic reporting

Line 52-56: were these studies conducted in China or across the world? Please clarify and point it out.

Line 60 - 62: It would be good to give a definition of and contrast schizophrenia vs. mental illness, in order for the audience not in the immediate field to understand the differences. A general description of schizophrenia would be very helpful.

Line 187: I suggest either reformatting or adding grids for the tables to improve readability.

Line 197: I suggest authors make an effort to use data visualization to tell the stories instead of just showing these tables. For example, consider using pie chart/bar chart or combinations.

Experimental design

Line 219: For completeness of the results, all factors and their OR/CI/P value should be reported. It can be included in the supplementary. Also, consider using forest plot to show these plots.

Line 270: It would be good to point to what is the suicide rate for the general population for comparison.

Validity of the findings

Supplementary file 1 raw data: The full names for each of the abbreviations column names should be provided (perhaps in a separate file)

·

Basic reporting

1) It is suggested that the statistical values of the full text be in a unified format and in italics.
2) The line 175 and line 197 should indent.
3) The layout of the table 1 & 2 is not clear enough.
4) Table 3 shows a multivariate logistic regression for correlates of disruptive behaviors (Line 219), but the title of Table 3 is “Multivariate logistic regression for correlates of violence”. Please check it again.

Experimental design

1. Regarding the inclusion and exclusion criteria:
1) There are two exclusion criteria (4), please check again. Besides, why exclude the people who absent for five medicine refills?
2) In inclusion criteria (4), it mentioned that living with a family member. If the patient lives with more than one family members, could he be able to attend the study?
3) In inclusion criteria (5), it mentioned that speaking Chinese. But it would cause bias because many patients aren’t able to attend the study who speak dialect only.
4) The exclusion criteria are effective supplements to the inclusion criteria, not repetition of the opposite of the inclusion criteria. Please check the criteria as follows.
① The inclusion criteria (3) vs the exclusion criteria (4)
② The inclusion criteria (4) vs the exclusion criteria (1)
③ The inclusion criteria (6) vs the exclusion criteria (3)
2. Regarding the Instruments:
1) You defined physical violence as any form of aggressive behavior toward other people such as scolding, cursing, scratching, punching, and hitting peoples. However, scolding and cursing should be categorized to language violence than physical violence.
2) The GAD-7 scale score ranges from 0 to 21, not from 0 to 27. Please check it again. (Spitzer RL, Kroenke K, Williams JB, Löwe B. A brief measure for assessing generalized anxiety disorder: the GAD-7[J]. Arch Intern Med, 2006, 166(10): 1092-1097.)
3. This cross-sectional study was conducted in Changsha, but the title of this paper describe it is in China. Please add the relevant description in the part of limitations.
4. In the discussion, it’ s better to add the innovation of the study.
5. There are a lot of types in the schizophrenia, such as “catatonic schizophrenia” which the manifestation is catatonic schizophrenia. I doubt if these types are comparable?

Validity of the findings

no comment

---

## Round 0.2 · accepted · Accept

Thanks for making all the necessary changes.